# Restricting Branched-Chain Amino Acids within a High-Fat Diet Prevents Obesity

**DOI:** 10.3390/metabo12040334

**Published:** 2022-04-07

**Authors:** Ming Liu, Yiheng Huang, Hongwei Zhang, Dawn Aitken, Michael C. Nevitt, Jason S. Rockel, Jean-Pierre Pelletier, Cora E. Lewis, James Torner, Yoga Raja Rampersaud, Anthony V. Perruccio, Nizar N. Mahomed, Andrew Furey, Edward W. Randell, Proton Rahman, Guang Sun, Johanne Martel-Pelletier, Mohit Kapoor, Graeme Jones, David Felson, Dake Qi, Guangju Zhai

**Affiliations:** 1Division of Biomedical Sciences (Genetics), Faculty of Medicine, Memorial University of Newfoundland, St. John’s, NL A1B 3V6, Canada; ming.liu@med.mun.ca; 2College of Pharmacy, University of Manitoba, Winnipeg, MB R3E 0T5, Canada; huangy49@myumanitoba.ca (Y.H.); dake.qi@umanitoba.ca (D.Q.); 3Discipline of Medicine, Faculty of Medicine, Memorial University of Newfoundland, St. John’s, NL A1B 3V6, Canada; hongwei8888@hotmail.com (H.Z.); prahman@mun.ca (P.R.); gsun@mun.ca (G.S.); 4Menzies Institute for Medical Research, University of Tasmania, Hobart, TAS 7000, Australia; dawn.aitken@utas.edu.au (D.A.); graeme.jones@utas.edu.au (G.J.); 5Department of Epidemiology and Biostatistics, University of California, San Francisco, CA 94158, USA; mnevitt@psg.ucsf.edu; 6Osteoarthritis Research Program, Division of Orthopaedics, Schroeder Arthritis Institute, University Health Network, Toronto, ON M5T 2S8, Canada; jason.rockel@uhnresearch.ca (J.S.R.); raja.rampersaud@uhn.ca (Y.R.R.); anthony.perruccio@uhnresearch.ca (A.V.P.); nizar.mahomed@uhn.ca (N.N.M.); mohit.kapoor@uhnresearch.ca (M.K.); 7Osteoarthritis Research Unit, University of Montreal Hospital Research Centre (CRCHUM), Montreal, QC H2X 0A9, Canada; dr@jppelletier.ca (J.-P.P.); jm@martelpelletier.ca (J.M.-P.); 8Department of Epidemiology, University of Alabama, Birmingham, AL 35233, USA; bethlew@uab.edu; 9Department of Epidemiology, University of Iowa, Iowa City, IA 52242, USA; james-torner@uiowa.edu; 10Institute of Health Policy, Management and Evaluation, Dalla Lana School of Public Health, University of Toronto, Toronto, ON M5T 2S8, Canada; 11Department of Surgery, University of Toronto, Toronto, ON M5T 2S8, Canada; 12Discipline of Surgery, Faculty of Medicine, Memorial University of Newfoundland, St. John’s, NL A1B 3V6, Canada; afurey99@gmail.com; 13Office of the Premier, Government of Newfoundland and Labrador, St. John’s, NL A1B 4J6, Canada; 14Discipline of Laboratory Medicine, Faculty of Medicine, Memorial University of Newfoundland, St. John’s, NL A1B 3V6, Canada; ed.randell@easternhealth.ca; 15Department of Rheumatology, Boston University School of Medicine, Boston, MA 02118, USA; dfelson@bu.edu; 16NIHR Manchester Biomedical Research Centre, Manchester Academic Health Science Centre, Manchester University NHS Foundation Trust, Manchester M13 9WL, UK

**Keywords:** obesity, metabolomics, meta-analysis, branched-chain amino acids, phenylalanine, tryptophan

## Abstract

Obesity is a global pandemic, but there is yet no effective measure to control it. Recent metabolomics studies have identified a signature of altered amino acid profiles to be associated with obesity, but it is unclear whether these findings have actionable clinical potential. The aims of this study were to reveal the metabolic alterations of obesity and to explore potential strategies to mitigate obesity. We performed targeted metabolomic profiling of the plasma/serum samples collected from six independent cohorts and conducted an individual data meta-analysis of metabolomics for body mass index (BMI) and obesity. Based on the findings, we hypothesized that restriction of branched-chain amino acids (BCAAs), phenylalanine, or tryptophan may prevent obesity and tested our hypothesis in a dietary restriction trial with eight groups of 4-week-old male C57BL/6J mice (n = 5/group) on eight different types of diets, respectively, for 16 weeks. A total of 3397 individuals were included in the meta-analysis. The mean BMI was 30.7 ± 6.1 kg/m^2^, and 49% of participants were obese. Fifty-eight metabolites were associated with BMI and obesity (all *p* ≤ 2.58 × 10^−4^), linked to alterations of the BCAA, phenylalanine, tryptophan, and phospholipid metabolic pathways. The restriction of BCAAs within a high-fat diet (HFD) maintained the mice’s weight, fat and lean volume, subcutaneous and visceral adipose tissue weight, and serum glucose and insulin at levels similar to those in the standard chow group, and prevented obesity, adipocyte hypertrophy, adipose inflammation, and insulin resistance induced by HFD. Our data suggest that four metabolic pathways, BCAA, phenylalanine, tryptophan, and phospholipid metabolic pathways, are altered in obesity and restriction of BCAAs within a HFD can prevent the development of obesity and insulin resistance in mice, providing a promising strategy to potentially mitigate diet-induced obesity.

## 1. Introduction

Obesity is a global pandemic with more than 1.9 billion adults overweight worldwide and over 650 million obese [1]. Obesity is strongly associated with the development or aggravation of multiple comorbidities, such as diabetes mellitus [2], cardiovascular disease [2], osteoarthritis (OA) [3], and others [4,5,6]. Obesity-related comorbidities are causing serious economic burden at all levels [7,8]. Public health and healthcare systems continue to face significant challenges in preventing and mitigating the negative health and socioeconomic consequences of the obesity pandemic. This is partly because metabolic alteration in obesity is not well elucidated.

Metabolomics, the comprehensive profiling of small molecule metabolites in a biological system, measures biochemical phenotypes of upstream genetic variation and downstream environmental influence, thus providing a highly integrated profile of biological status [9]. The application of metabolomics has led to the discovery of metabolic biomarkers associated with cardiovascular disease [10], neurological disease [11], cancer [12], osteoarthritis [13], as well as COVID-19 [14]. Recent metabolomics studies have identified a signature of altered amino acid profiles highlighting branched-chain amino acids (BCAAs) to be associated with obesity [15] and the degree of obesity [16]; however, it is not clear how these alterations are implicated in the development of obesity, and whether these findings have actionable clinical potential. Additionally, the aforementioned studies interrogated single cohort with small sample size and lacked independent validation. Therefore, we performed an individual data meta-analysis of metabolomics on body mass index (BMI) and obesity in 3397 study participants from six cohorts collected from Canada, Australia, and USA. Based on the meta-analysis results, we performed a dietary restriction trial in mice and demonstrated that the restriction of BCAAs within a high fat diet (HFD) could prevent the development of obesity and insulin resistance, providing a promising strategy to conquer the global pandemic of obesity.

## 2. Results

A total of 3397 individuals were included in the study, of which 58% were female. A total of 91% of the participants were Caucasian. The mean age was 62.6 ± 9.4 years, and the mean BMI was 30.7 ± 6.1 kg/m^2^, with 49% of participants obese, and 16% having normal weight. The characteristics of all six cohorts included in this study are presented in Table 1. 

### 2.1. BMI/Obesity Associated Metabolites and Pathways

Sixty-six (66) metabolites were found to be significantly associated with BMI (*p* ≤ 1.54 × 10^−4^, Appendix A) and 63 were significantly associated with obesity (*p* ≤ 3.77 × 10^−4^, Appendix A) after adjusting for age, sex and OA status. The meta-analysis results for BMI are presented by the volcano plot of Figure 1A and those for obesity are presented by the volcano plot of Figure 1B. Fifty-eight (58) metabolites were significantly associated with BMI and obesity (Figure 1C) after adjusting for age, sex and OA status, of which 10 were amino acids, 2 were acylcarnitines, and 46 were phospholipids. 

The forest plots of Figure 2A present the results of the most significant amino acids for each cohort and the meta-analysis results for BMI, and the forest plots of Figure 2B present those for obesity.

The 10 amino acids and 2 acylcarnitines significantly associated with both BMI and obesity were manually mapped into KEGG metabolic pathways. Seven out of the ten amino acids, isoleucine, valine, glutamate, alanine, glycine, serine, and asparagine, as well as C3 acylcarnitine were mapped into BCAA metabolic pathway (Figure 3), highlighting the importance of the BCAA catabolic pathway in obesity. Phenylalanine and tyrosine were mapped into the phenylalanine degradation pathway (Figure 3), and kynurenine was mapped into the tryptophan metabolic pathway (Figure 3). 

Concentrations of all identified phosphatidylcholines (PCs) were lower in obese subjects or negatively associated with BMI except for one, indicating either biosynthesis of PCs was reduced, or their degradation was increased in obesity (Appendix A). 

### 2.2. Relationships between BMI and Obesity Associated Metabolites and Diabetes, and Levels of Serum Glucose, Insulin and Homeostatic Model Assessment for Insulin Resistance (HOMA-IR)

Among the 58 metabolites significantly associated with both BMI and obesity, isoleucine was positively associated with diabetes in the Complex Diseases in the Newfoundland population: Environment and Genetics (CODING), Longitudinal Evaluation in the Arthritis Program: Osteoarthritis Study (LEAP-OA), the Newfoundland Osteoarthritis Study (NFOAS), and the Tasmanian Older Adult Cohort study (TASOAC) (*p* ≤ 0.04, Appendix A), and positively associated with glucose, insulin, and HOMA-IR levels in the CODING cohort (*p* ≤ 0.01, Appendix A). These results suggested that BCAA metabolic pathways were associated with not only obesity, but also diabetes and insulin resistance.

### 2.3. Relationships between BMI and Obesity Associated Metabolites and Blood Lipids Profile in the CODING and NFOAS Cohorts

Six metabolites positively associated with BMI and obesity, valine, isoleucine, glutamate, alanine, phenylalanine, and C3, were all negatively correlated with high-density lipoprotein (HDL) cholesterol (*p* ≤ 0.02, Appendix A), while positively correlated with very-low-density lipoprotein (VLDL) and TG in both cohorts (both *p* ≤ 0.01, Appendix A). In addition, valine, isoleucine, glutamate, alanine, phenylalanine, and tyrosine were all positively correlated with total cholesterol (TC)/HDL ratio in both cohorts (*p* ≤ 0.02, Appendix A), indicating that BCAA and phenylalanine and tyrosine metabolic pathways could be implicated in the cardiovascular diseases that are highly prevalent among obese individuals [17]. Blood lipid levels in obese and non-obese groups in these two cohorts are presented in Appendix A.

### 2.4. Relationships between BMI and Oobesity Associated Metabolites and Visceral Fat Mass in the CODING Cohort

Valine, isoleucine, glutamate, alanine, kynurenine, tyrosine, C0, C3, and PC aa C38:3 were all positively correlated with visceral fat mass. In contrast, glycine, serine, asparagine and 34 phospholipid species were negatively correlated (all *p* ≤ 0.04, Appendix A), mirroring the relationships of these metabolites with BMI and obesity found in the meta-analysis. Thus, these results suggested that the metabolic pathways associated with BMI and obesity were likely behind some disorders associated with visceral fat mass as well.

### 2.5. Amino Acids Restriction and Obesity in Mice

To further define the roles of BCAA, phenylalanine and tryptophan pathways in regulating the development of obesity and insulin resistance, we conducted a feeding experiment with or without amino acid restriction. HFD induced obesity (Figure 4A) and increased fat accumulation (Figure 4B–D and Appendix A) and adipocyte hypertrophy in both subcutaneous and visceral fat (Figure 4E) which were completely prevented by restriction of BCAAs (Figure 4A–E). Further restriction of phenylalanine and tryptophan had no further effects (Figure 4A–E) or attenuated the effect of BCAA restriction on visceral fat accumulation (Figure 4D).

Obesity is linked with chronic inflammation in adipose tissue and is associated with an imbalance in the ratio of M1 (proinflammatory)/M2 (anti-inflammatory) macrophages. We observed that HFD significantly upregulated M1Φ while downregulating M2Φ, leading to an imbalance in the ratio of M1/M2, which was prevented by the restriction of BCAAs (Figure 4F–H). Interestingly, restricting BCAAs per se had the strongest effect on preventing the decrease of the M2Φ level (Figure 4F,G). Adipose inflammation contributes to the development of whole-body insulin resistance. We also found increases in serum glucose, insulin (Figure 4I,J), fatty acid (Appendix A) levels, lipid deposition in liver (Appendix A), and impaired insulin sensitivity (Figure 4K) in the HFD group. The restriction of BCAAs in HFD prevented these metabolic dysfunctions (Figure 4I,J) and maintained insulin sensitivity (Figure 4K). Additional restriction of phenylalanine and tryptophan had no further effects (Figure 4I–K).

## 3. Discussion

To the best of our knowledge, this was the first individual data meta-analysis of metabolomics on BMI and obesity in populations from Canada, Australia, and USA. Seven out of the ten BMI/obesity-associated amino acids and one of the two acylcarnitines were mapped into the BCAA catabolic pathway, highlighting the importance of BCAA metabolism in obesity. In addition, the alterations of phenylalanine, tryptophan and phospholipid metabolic pathways were found to be associated with BMI and obesity. Although we could not infer any causal relationships between these altered metabolic profiles and obesity from the meta-analysis, the results helped to generate novel hypothesis to be tested in our dietary restriction trial in mice, which demonstrated that restriction of BCAAs within a HFD could prevent the development of obesity, adipose inflammation, elevation in serum glucose and insulin levels, and insulin resistance induced by HFD, providing a promising strategy to mitigate diet-induced obesity.

In cytosol and mitochondria, the amino groups of BCAAs are transferred to α-ketoglutarate (α-KG) to form glutamate while the corresponding branched-chain keto acids (BCKAs) are formed [18,19]. Glutamate acts as an amino group source to form alanine from pyruvate, using glycine as a carbon source via its sequential conversion to serine and pyruvate, a step directly influenced by BCAAs supply [20]. Glutamate also forms aspartate via transamination of oxaloacetate, and forms glutamine via amidation. Glutamine transported into mitochondria is converted to glutamate, which can be converted to α-KG to enter the tricarboxylic acid (TCA) cycle, or to ornithine and then citrulline, which is then transported back into cytosol for arginine synthesis. All these AAs can be released into the blood and then distributed to all the body tissues. Upon import into the mitochondrial matrix, BCKAs undergo irreversible decarboxylation by BCKA dehydrogenase (BCKD) to form corresponding acyl-CoA esters—leucine to acetyl-CoA; isoleucine and valine to propionyl CoA and then to succinyl-CoA. Both acetyl-CoA and succinyl-CoA can then be incorporated into the TCA cycle. When in excess, propionyl CoA is transported to the cytoplasm and plasma as C3 acylcarnitine by the combined action of carnitine palmitoyl transferase II (CPT2) and carnitine-acylcarnitine translocase (CACT) [21].

Elevated BCAA levels have been consistently observed in obesity [15,22]. When BCAAs accumulate in plasma, the flux of these amino acids into skeletal muscle and liver and their catabolism are increased, resulting in increased production of propionyl CoA and succinyl CoA. These metabolic intermediates replenish the TCA cycle and contribute to the accumulation of incompletely oxidized intermediates of fatty acid oxidation, which in turn reduces the efficiency of glucose oxidation, leading to mitochondrial stress, impaired insulin action, and ultimately perturbation of glucose homeostasis [23]. Excess fatty acids are then deposited into adipose tissue, where fatty acids are esterified to glycerol and form energy-rich TG, which eventually leads to adipocyte hypertrophy. Alternatively, high levels of propionyl CoA and succinyl CoA could promote malonylation and succinylation of key metabolic enzymes in mitochondrial and disturb fatty acid and glucose oxidation processes [24]. Another link between BCAA catabolism and obesity is acetyl-CoA, a central metabolic intermediate involved in energy metabolism and cell homeostasis. Increased BCAAs lead to the overproduction of acetyl-CoA from BCKA oxidation in mitochondria. In addition to entering TCA cycle, the overabundant acetyl-CoA can be transported to cytosol in the form of citrate which exerts a negative feedback on glycolysis by binding to a glycolytic enzyme phosphofructokinase (PFK), and promotes gluconeogenesis and lipid synthesis by activating fructose-1,6-bisphosphatase 1 (Fbp1) and acetyl-CoA carboxylase alpha (ACACA). Citrate can be converted back to acetyl-CoA, which then acts as the precursor for the production of fatty acids, cholesterol, PCs and lysoPCs [25,26], which are all implicated in obesity and insulin resistance [27,28,29]. Acetyl-CoA is also the sole donor of the acetyl groups for protein acetylation [25], which plays a pivotal role in fatty acid and glucose metabolism [30]. 

Increased circulating concentrations of phenylalanine and tyrosine have often been reported in obesity [15,31]. Phenylalanine is an essential amino acid and the precursor of tyrosine. Tyrosine is further metabolized to acetoacetate and fumarate to enter TCA cycle, also can be converted to 3,4-dihydroxyphenylalanine (L-DOPA) by tyrosine hydroxylase (TH), which is rate-limiting for the subsequent synthesis of the catecholamines including dopamine, norepinephrine, and epinephrine [32]. Similar to BCAAs, the elevated phenylalanine and tyrosine levels in obesity could generate overabundant metabolic intermediates to enter TCA cycle. In addition, it could promote an increase in blood glucose level through the anti-insulin action of catecholamines [33].

Tryptophan and kynurenine have also been reported to be positively associated with obesity [33]. Most of the essential amino acid tryptophan (>95% in mammals) is catabolized through the kynurenine pathway, producing kynurenine which is further metabolized along three distinct routes to form quinolinate, kynurenate, and xanthurenate, generating glutamate from α-KG, a key molecule in the TCA cycle. A small percentage of tryptophan is hydroxylated to synthesize serotonin and melatonin. Kynurenine levels were increased in the obesity group in our study, indicating the upregulation of kynurenine pathway which could lead to the overproduction of xanthurenic acid. This compound has been proposed to be one of the factors predisposing to insulin resistance [34]. 

BCAAs, phenylalanine, tyrosine and tryptophan are all large neutral amino acids (LNAAs) that compete for uptake into tissues through the shared LNAA transporter 1 (LAT1) [15], which could in part explain the observed elevation of all these amino acids in obese individuals. Given that all these amino acids are essential, we investigated their potential clinical actionability in mice through a dietary restriction trial. We demonstrated that a HFD induced obesity and impaired insulin sensitivity in mice, but these detrimental effects were prevented by restricting BCAAs within a HFD. 

Specifically, we demonstrated that BCAA restriction prevented HFD-augmented excessive weight gain and adipose tissue accumulation as well as adipocyte hypertrophy, also lowered fatty acid levels. In addition, BCAA restriction within a HFD helped maintain normal glucose and insulin levels and prevented adipose inflammation and insulin resistance. However, additional dietary restriction of phenylalanine and tryptophan had no further effects or somehow attenuated the effects of BCAAs restriction. This may suggest that the alteration of phenylalanine and tryptophan pathways in obesity is not independent but secondary to the alteration of the BCAA catabolism. Interestingly, we did not observe such an effect when restricting these amino acids within a SCD except for the limited weight gain when all these amino acids were restricted together within a SCD (Appendix A). Previous studies have shown that the metabolic health restoration by BCAA restriction was not mediated by caloric restriction or increased activity [35,36], so the shifting of metabolic processes is likely involved. Given the competing nature of BCAAs and fatty acids to enter TCA cycle for energy production and BCAAs are a preferred fuel for muscle, our results appeared to support a model in which under a HFD condition, restriction of BCAAs reduces metabolic intermediates (such as acetyl-CoA and succinyl CoA) entering TCA cycle, relieves mitochondrial overloading caused by high levels of fatty acids, and thus facilitates efficient and complete fatty acid oxidation. This in turn helps maintain glucose homeostasis and keep the body in a balanced energy mode. This hypothesis was supported by the normal glucose, insulin, and fatty acid levels and fat volume in the HFD-BCAA group observed in the current and a few other studies [35,36]. We also found that BCAA restriction within a HFD prevented the decrease of M2Φ as seen in HFD group. The polarization of M2Φ requires both fatty acid oxidation and glycolysis [37]. This result further suggested that BCAA restriction promoted these two processes, and could subsequently prevent chronic inflammation linked to obesity. In addition, the beneficial effects of BCAA restriction could also be achieved via the regulation of genes related to energy production or balance. Cummings et al. and Karusheva et al. reported that BCAA restriction suppressed the expression of numerous lipogenic genes and induced the expression of fibroblast growth factor 21 (FGF21) [35,36,38], a hormone that regulates energy balance and glucose and lipid homeostasis through a heterodimeric receptor complex comprising FGF receptor 1 (FGFR1) and β-klotho [39]. Administration of FGF21 to rodents or non-human primates causes considerable pharmacological benefits on a cluster of obesity-related metabolic complications including a reduction in fat mass and alleviation of hyperglycemia, insulin resistance, and dyslipidemia [39]. FGF21 exerts these effects by increasing whole body energy expenditure (driven by brown adipose tissue thermogenesis), increasing secretion of adiponectin, a protein hormone and adipokine involved in regulating glucose levels, fatty acid breakdown, and M2Φ polarization, and influencing central reward circuits [40]. Whether the metabolic alterations observed after administration of BCAA restriction were mediated partially by FGF21 remains to be further investigated. It was also possible that when BCAAs were restricted, reduced acetyl-CoA production from BCAA oxidation prevented the negative feedback on glycolysis and promotion of gluconeogenesis and lipid synthesis mediated by citrate in cytosol. As previously mentioned, acetyl-CoA is a key node in metabolism, and when acetyl-CoA concentration changes, subcellular compartmentalization and the fate of acetyl-CoA undergo a major shift—high nucleocytosolic acetyl-CoA level directs acetyl-CoA away from the mitochondria and back to the cytosol, promoting its utilization for lipid synthesis, while lower nucleocytosolic acetyl-CoA level shifts metabolism towards oxidation of acetyl-CoA in the mitochondria for ATP synthesis, and limits fatty acid synthesis and other growth-related processes [41]. 

In agreement with our findings, Deyang et al. [36] recently reported that reduction of isoleucine or valine, or all three BCAAs together in a high-fat, high-sucrose western diet (WD) restored the metabolic health of diet-induced obese (DIO) mice. Fontana et al. [42] and Deyang et al. [36] reported that the reduction of BCAAs in a protein restricted diet with normal dietary fat level was sufficient to improve glucose tolerance and body composition equivalently to a diet reduced in all amino acids. Cummings et al. [35] showed that switching to an extra low BCAA diet from WD could restore metabolic health in the WD-induced obese mice. McGarrah et al. [43] reported that feeding a BCAA-restricted chow diet to Zucker fatty rats caused a shift in cardiac fuel metabolism that favored fatty acid relative to glucose catabolism. While these studies had different study designs and addressed different questions, their results support our findings. Zhou et al. [44] recently reported that upregulating BCAA catabolism by inhibiting BCKD kinase (BCKDK), an enzyme that phosphorylates and inhibits BCKD, could improve insulin sensitivity in genetically induced obese mice, but this model may not actually mimic obesity in human (the BCAA levels were not elevated in these mice). Although they also showed that BCKDK inhibitor can increase BCKD activity, reduce plasma levels of BCAAs and BCKAs, and improve insulin sensitivity in DIO mice, this intervention did not reduce obesity. This appears to suggest that reducing BCAA intake rather than increasing their catabolism has a better potential in preventing obesity. 

The strength of the current study included individual data meta-analysis of metabolomics with a large sample size and a dietary restriction trial in mice. It was noted that in meta-analysis, the degree of heterogeneity was highly variable which cannot be explained by the different cohorts included in the study. Hence, we used the random-effect model to take the heterogeneity into account. This study also had some limitations. Firstly, we were unable to examine whether the participants included in this study were representative of the entire cohorts due to the lack of access to information of participants without metabolomics data. However, the original metabolomic study of each cohort was not for obesity, thus we wouldn’t expect a potential selection bias in relation to obesity. In addition, 9% of the study participants were non-Caucasian from multiple ethnicities with very small numbers of participants from certain ethnicities, and the classifications of ethnicity were different in different cohorts included in the study, hence we were unable to analyze the influence of ethnicity differences on BCAA metabolic pathway. Additionally, the study participants were older and largely affected by OA, which could limit the application of our findings to general population. However, the OA status was adjusted in data analysis, and our mice experiment strongly supported the findings of the population meta-analysis, suggesting this may not be an issue. Secondly, we did not have data on parameters such as smoking status/history, alcohol consumption, physical activity, medications, diet, etc., which could be associated with BMI and/or affect metabolomics profiles; further studies with adjustment of these parameters are needed to confirm our findings. Diabetes status was self-reported in all cohorts, and we had no data on whether these participants had type I diabetes, which could have increased the possibility of misidentification of the association between metabolites and diabetes. In mice experiment, only male mice were utilized in the dietary restriction trial, which may limit the generalizability of our findings to females. Lastly, the clinical relevance of the findings needs to be confirmed in humans. Although the sample size was very small with only 12 individuals, a very recent study in humans [45] showed that a BCAA-restricted diet marginally lowered HOMA-IR levels, suggesting the potential of dietary BCAA restriction as a novel nutraceutical modality in controlling obesity and potentially preventing obesity related medical illness.

## 4. Materials and Methods

### 4.1. Study Participants

Six independent cohorts were included in the study: CODING [13,46], LEAP-OA [47], the licofelone/naproxen clinical trial cohort [48,49,50], NFOAS [13,51,52], TASOAC [13,53], and the Multicenter Osteoarthritis Study (MOST) [54,55] cohorts. All human studies obtained ethics approval and written informed consent was obtained from all study participants. Detailed information on individual studies can be found in the Appendix A.

### 4.2. Metabolomics Profiling

Fasting plasma or serum was separated from the whole blood and stored at −80 °C until analysis. The Biocrates AbsoluteIDQ^®^ p180 kit (BIOCRATES Life Sciences AG, Innsbruck, Austria) was utilized for metabolomic profiling in all cohorts except for 409 samples of the TASOAC, which were done by the TMIC Prime Metabolomics Profiling Assay. One hundred fourteen metabolites overlapped between the two assays. Our in-house reproducibility of the p180 assay was performed in 23 samples as previously described [56], and the mean coefficient of variation (CV) for all metabolites was 0.07 ± 0.05 µM. The CV for the metabolites detected with the TMIC assay kit ranged between 1.16–15.93% [53]. Details of the assays and profiling methods [56,57] are provided in the Appendix A, and the full lists of the metabolites quantified by these two assay kits are provided in Appendix A.

### 4.3. Demographic Data Collection and Biochemical Parameter Measurement

Data on age, sex, weight, height and ethnicity were available for all cohorts. Data on self-reported diabetes status was available for all cohorts except for MOST. Data on blood lipid profiles were available for CODING and NFOAS cohorts [58]. Levels of glucose, insulin, and HOMA-IR as well as visceral fat mass were measured in the CODING cohort [59]. Details of the methods are provided in the Appendix A.

### 4.4. Experimental Animals and Dietary Regimens

Forty 4-week male C57BL/6J mice were randomly divided into eight groups and fed with (1) standard chow diet (SCD, A20101503); (2) SCD with 2/3 reduction of BCAAs (SCD-BCAAs, A20120804); (3) SCD with 2/3 reduction of BCAAs and phenylalanine (SCD-BCAAs-Phe, A20120805); (4) SCD with 2/3 reduction of BCAAs, phenylalanine, and tryptophan (SCD-BCAAs-Phe-Trp, A20120806); (5) high-fat diet (HFD, A20101501); (6) HFD with 2/3 reduction of BCAAs (HFD-BCAAs, A20120801); (7) HFD with 2/3 reduction of BCAAs and phenylalanine (HFD-BCAAs-Phe, A20120802); and (8) HFD with 2/3 reduction of BCAAs, phenylalanine, and tryptophan (HFD-BCAAs-Phe-Trp, A20120803), respectively (Appendix A), for 16 weeks. All experiments involving mice followed ethical guidelines (details in the Appendix A).

### 4.5. Metabolic Assessment, Body Composition Evaluation, and Inflammation Marker Measurement in Mice

Following feeding experiments, intraperitoneal glucose tolerance test (IPGTT) and insulin tolerance test (IPITT) were performed after 8 h of fasting. Serum levels of glucose, insulin, TG, and free fatty acid were quantified. Body compositions including fat and lean volume percentages were evaluated by a micro-CT system. Inguinal subcutaneous white adipose tissue (iWAT) and epidydimal white adipose tissue (eWAT) were collected for tissue weight measurement and Hematoxylin and Eosin (H&E) staining (iWAT and eWAT). Stromal vascular fraction (SVF) was isolated from eWAT and labelled with CD45, F4/80, CD11b, CD301, and CD11c to identify M1 and M2 macrophages (Appendix A). Details of the methods are provided in the Appendix A.

### 4.6. Statistical Analysis

Risk of bias assessment for studies included in the meta-analysis were performed [60] (Appendix A). One hundred twenty-nine (129) metabolites passed quality control (see Appendix A) and a two-stage meta-analysis was then performed. First, BMI and obesity (obese defined by BMI ≥ 30 kg/m^2^ and non-obese defined by BMI < 25 kg/m^2^) were regressed on each standardized metabolite concentrations with adjustment for age, sex, and OA status in each cohort [61]. Then random effects meta-analysis with inverse variance as weights was performed on the summary statistics for each cohort. The significance level for metabolites associated with BMI/obesity was defined at α = 0.0004 after controlling multiple testing of 129 metabolites with the Bonferroni method. Next, associations between standardized metabolite concentrations and diabetes status were assessed using multivariable logistic regression adjusting for age and sex. Associations between standardized metabolite concentrations and levels of blood lipids, glucose, insulin, HOMA-IR, and visceral fat mass were assessed using Spearman’s rank correlation. Student’s *t*-test, one-way analysis of variance (ANOVA) with Tukey’s post hoc tests or multivariate (two-way) ANOVA were utilized for all the variables generated in the mice feeding experiment including body weight, fat and lean volume, adipose tissue weight, serum glucose and insulin concentrations, and M1 and M2 macrophage counts. The significance level was defined at α = 0.05 for all data analyses in mice. All analyses were performed in R Studio with R version 3.6.3 and Prism. Visualizations of the results were done with ggplot2, VennDiagram and Forestplot R packages [62] and Prism.

## 5. Conclusions

In conclusion, our data suggested that obesity was associated with altered metabolic pathways including BCAA, phenylalanine, tryptophan, and phospholipid metabolic pathways; restricting BCAAs within an HFD can prevent the development of obesity and insulin resistance in mice. While a clinical trial in humans is needed, our findings provided a promising strategy to potentially mitigate diet-induced obesity.

## Figures and Tables

**Figure 1 metabolites-12-00334-f001:**
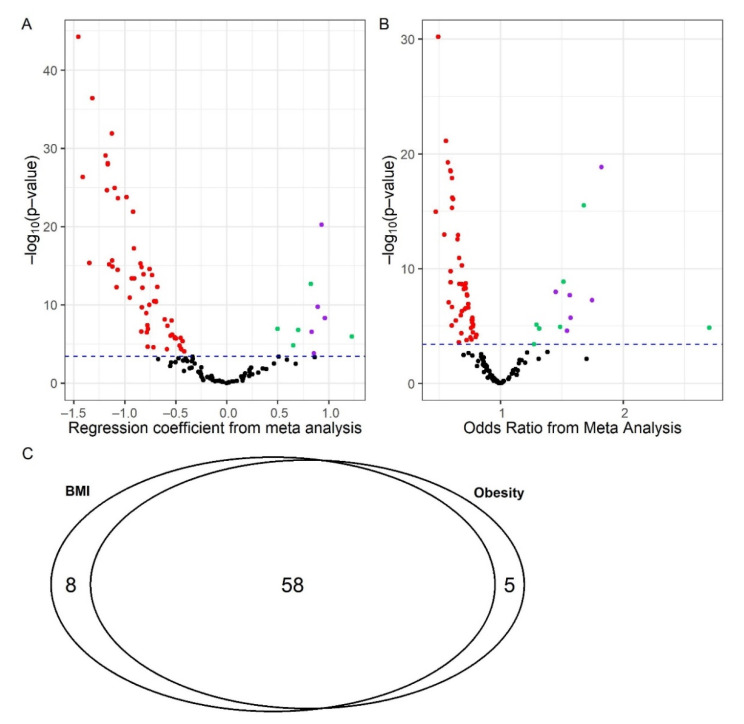
Volcano plots and Venn diagram for metabolites significantly associated with BMI and Obesity. (**A**) Metabolites significantly associated with BMI; (**B**) metabolites significantly associated with obesity; (**C**) number of metabolites associated with both BMI and obesity. BMI: body mass index. (**A**,**B**) Red indicates metabolites significantly negatively associated with body mass index or obesity; green indicates metabolites significantly positively associated with body mass index or obesity; black indicates metabolites not significantly associated with body mass index or obesity; purple indicates major amino acids significantly associated with body mass index or obesity which are presented in the forest plots in Figure 2.

**Figure 2 metabolites-12-00334-f002:**
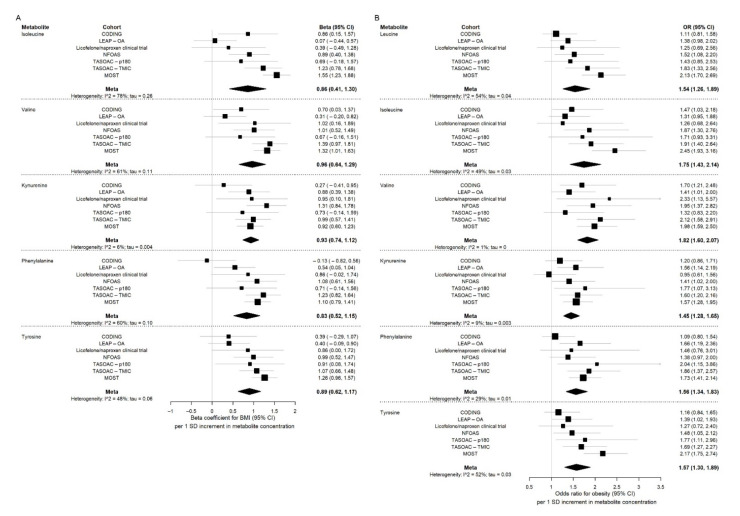
Forest plots for major amino acids associated with BMI or obesity. (**A**) Forest plots for major amino acids associated with BMI; (**B**) forest plots for major amino acids associated with obesity. BMI: body mass index; CODING: Complex Diseases in the Newfoundland population: Environment and Genetics; LEAP-OA: Longitudinal Evaluation in the Arthritis Program, Osteoarthritis Study; NFOAS: Newfoundland Osteoarthritis Study; TASOAC-p180: Tasmanian Older Adult Cohort profiled with Biocrates AbsoluteIDQ^®^ p180 kit; TASOAC-TMIC: Tasmanian Older Adult Cohort profiled with the TMIC Prime Metabolomics Profiling Assay; MOST: Multicenter Osteoarthritis Study; OR: odds ratio; CI: confidence interval; SD: standard error. Beta coefficient and 95% CI for individual cohorts were obtained by multivariable linear regression adjusting for age, sex, and osteoarthritis status; OR and 95% CI for individual cohorts were obtained by multivariable logistic regression adjusting for age, sex, and osteoarthritis status; those for meta-analysis were obtained by random effects meta-analysis with inverse variance as weights on the summary statistics from each cohort. Square indicates individual study effect estimate, and the size of the square indicates study weight; diamond indicates the overall effect estimate from meta-analysis.

**Figure 3 metabolites-12-00334-f003:**
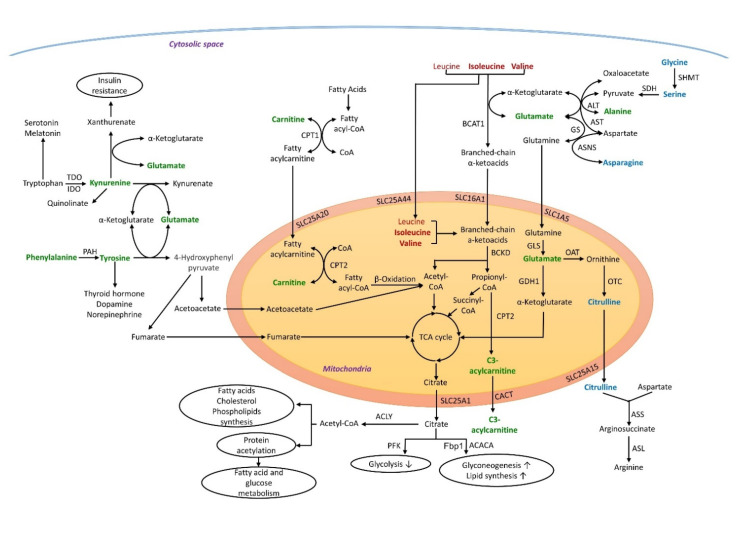
Metabolic pathways associated with BMI and obesity. Bold font indicates metabolites significantly associated with obesity. Red indicates branched-chain amino acids; blue indicates metabolites negatively associated with obesity; green indicates metabolites positively associated with obesity. ACACA: acetyl-CoA carboxylase alpha; ACLY: ATP-citrate lyase; ALT: alanine aminotransferase; ASL: argininosuccinate lyase; ASNS: asparagine synthetase; ASS: argininosuccinate synthase; AST: aspartate aminotransferase; BCAT: branched-chain amino acid aminotransferase; BCKD: branched-chain α-keto acid dehydrogenase; CACT: carnitine-acylcarnitine translocase; CoA: acyl-coenzyme A; CPT: carnitine palmitoyltransferase; Fbp1:1,6-bisphosphatase; GDH: glutamate dehydrogenase; GLS: glutaminase; GS: glutamine synthetase; IDO1: indoleamine 2,3-dioxygenase-1; OAT: ornithine aminotransferase; OTC: ornithine transcarbamylase; PAH: phenylalanine 4-hydroxylase; PFK: phosphofructokinase; SDH: serine dehydratase; SHMT: serine-hydroxymethyltransferase; SLC1A5: Solute Carrier Family 1 Member 5; SLC16A1: Solute Carrier Family 16 Member 1; SLC25A1: Solute Carrier Family 25 Member 1; SLC25A15: Solute Carrier Family 25 Member 15; SLC25A20: Solute Carrier Family 25 Member 20; TCA: tricarboxylic acid; TDO2: tryptophan 2,3-dioxygenase.

**Figure 4 metabolites-12-00334-f004:**
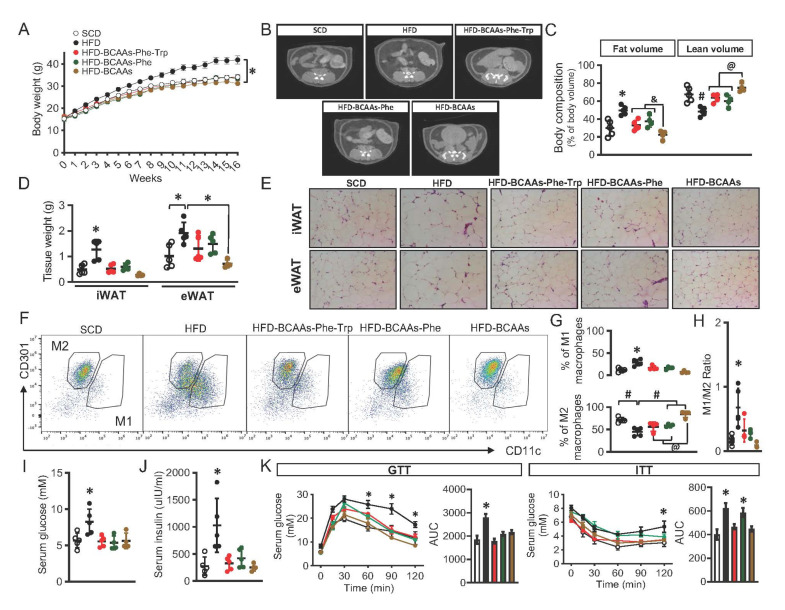
Metabolic regulation with restriction of BCAAs, phenylalanine and tryptophan. C57BL/6J mice (4 weeks) were fed by high-fat diet (HFD) with or without combinations of restrictions in branched-chain amino acids (BCAAs), phenylalanine and tryptophan for 16 weeks. Body weight was monitored weekly (**A**) and body composition was detected by a micro CT scanning at the end of experiment in (**B**,**C**). Adipose tissue weight (**D**) and Hematoxylin and Eosin (H&E) staining (**E**) were performed to further evaluate any alterations in adipose tissue. M1 and M2 macrophages (**F**–**H**) in isolated stromal vascular fraction (SVF) were quantified by a flow-cytometer. Fasting blood glucose and serum insulin levels were detected (**I**,**J**) and insulin resistance in these mice were measured by intraperitoneal glucose tolerance test (GTT) and insulin tolerance test (ITT) (**K**). All data are presented as mean ± standard deviation (SD). * *p* ≤ 0.05 increase vs. all other groups in ((**A**,**C**,**D**) left), (**G**) to (**K**) or vs. the groups of SCD and HFD-BCAAs in ((**D**) right). & *p* ≤ 0.05 reduction vs. both groups of HFD-BCAAs-Phe-Trp and HFD-BCAAs-Phe in (**C**). # *p* ≤ 0.05 reduction vs. all other groups or vs. the groups of SCD, HFD-BCAAs-Phe and HFD-BCAAs in (**C**,**G**). @ *p* ≤ 0.05 increase vs. the groups of HFD-BCAAs-Phe-Trp and HFD-BCAAs-Phe in (**C**,**G**). iWAT-inguinal subcutaneous white adipose tissue. eWAT- epidydimal white adipose tissue. BCAA: branched-chain amino acids; SCD: standard chow diet; HFD: high-fat diet; HFD-BCAA: high-fat diet with 2/3 reduction of branched-chain amino acids; HFD-BCAAs-Phe: high-fat diet with 2/3 reduction of branched-chain amino acids and phenylalanine; HFD-BCAAs-Phe-Trp: high-fat diet with 2/3 reduction of branched-chain amino acids, phenylalanine, and tryptophan. Color codes indicated on (**A**) are the same as for (**C**,**D**,**G**–**K**).

**Table 1 metabolites-12-00334-t001:** Characteristics of study participants.

Cohort	Population Location	N	Ethnicity (% of Caucasian)	Age (Years)	Sex (% of Females)	BMI (kg/m^2^)	Obese (%)	Normal Weight (%)	Prevalence of Diabetes (%)
CODING	Canada	226	100%	48.9 ± 12.7	60%	28.9 ± 5.1	45%	26%	15%
LEAP-OA	Canada	495	82%	65.5 ± 8.4	57%	30.8 ± 6.0	51%	15%	15%
Licofelone/naproxen clinical trial	Canada	158	98%	60.7 ± 8.0	69%	31.8 ± 5.7	59%	11%	8%
MOST	U.S.	1248	85%	61.8 ± 7.8	62%	30.6 ± 5.8	48%	15%	-
NFOAS	Canada	704	99%	65.4 ± 9.6	55%	33.3 ± 7.0	65%	8%	19%
TASOAC	Australia	566	98%	64.1 ± 6.6	52%	27.8 ± 4.7	27%	30%	2%
Total		3397	91%	62.6 ± 9.4	58%	30.7 ± 6.1	49%	16%	12%

CODING: Complex Diseases in the Newfoundland population: Environment and Genetics; LEAP-OA: Longitudinal Evaluation in the Arthritis Program, Osteoarthritis Study; MOST: Multicenter Osteoarthritis Study; NFOAS: Newfoundland Osteoarthritis Study; TASOAC: Tasmanian Older Adult Cohort; BMI: body mass index.

## Data Availability

The data presented in this study are available on request from the corresponding author. The data are not publicly available due to the restriction on the ethics approval.

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
