# Peer review of "Restricting Branched-Chain Amino Acids within a High-Fat Diet Prevents Obesity"

_metabolites, 2022, doi:10.3390/metabo12040334_

Round 1

Reviewer 1 Report

Ming Li and colleagues characterized the serum metabolites, correlated explicitly with obesity and body mass index (BMI) by meta-analysis of metabolomics. The authors hypothesized that restriction of branched-chain amino acids (BCAA) protects from obesity. The authors also conducted experiments to prove their hypothesis using mice fed with a high-fat diet without BCAA supplement. Surprisingly, mice without BCAA supplementation reduced their body weights. It seems that the author’s manuscript looks nice. But there are some questions to be answered.

Mouse experiments

  1. What is the effect of the reduction of BCAA supplements on skeletal muscles? How was skeletal muscle weight affected in the HFD-BCAAs group in Fig. 4A? Was skeletal muscle weight reduced in mice without BCAA supplementation?
  2. Based on the authors’ hypothesis, BCAA content should be increased in the HFD group. Did the authors measure BCAA content in sera of the HFD mice?
  3. The authors used juvenile mice for experiments, although human metabolic data were obtained from participants about 60 years old. Can the authors explain the reason not to use old mice? It would be better to use old obese mice.

Reviewer 2 Report

In this study the authors examine the metabolic alterations of obesity and to explore potential strategies to mitigate it.

Although the study has the potentiality of being shared with the scientific community, I believe that the manuscript would benefit from a minor revision with the attempt to better support their experimental setting.

  1. Abstract: they should start with a first paragraph describing the background. This section should outline the following information:
  • What is already known about the subject, related to the paper in question
  • What is not known about the subject and hence what the study intended to examine (or what the paper seeks to present)
  1. The theoretical framework is scarce, they should clearly describe the scientific evidence that supports the hypothesis they have raised.

  1. A lot of necessary information is missing in methods section:
  • Experimental procedures should be better defined
  • More information should be provided about the participants’ characteristics.
  • They should better defined inclusion and exclusion criteria

  1. The Discussion should be enriched with the existing theory. The authors should clearly describe the scientific evidence that supports their findings.

  1. I would like to see more of the practical implications. Based on the analyzed variables, how the authors intend to use their findings?

Kind regards

Reviewer 3 Report

In the present study, the authors performed targeted metabolomic profiling of the plasma/serum samples collected from six independent cohorts and conducted an individual data meta-analysis of metabolomics for body mass index (BMI) and obesity. Fifty-eight metabolites were associated with BMI and obesity, linked to alterations of the branched-chain amino acids (BCAAs), phenylalanine, tryptophan, and phospholipid metabolic pathways.

The restriction of BCAAs within a high-fat diet (HFD) maintained the mice’s weight, fat and lean volume, subcutaneous and visceral adipose tissue weight, and serum glucose and insulin at levels similar to those in the standard chow group, and prevented obesity, adipocyte hypertrophy, adipose inflammation, and insulin resistance induced by HFD.

The authors concluded that four metabolic pathways including BCAA, phenylalanine, tryptophan, and phospholipid metabolic pathways are altered in obesity and restriction of BCAAs within a HFD can prevent the development of obesity and insulin resistance in mice, providing a promising strategy to potentially mitigate diet-induced obesity. The findings are interesting however, I have several concerns.

  1. The restriction of BCAAs was used as HFD with 2/3 reduction of BCAAs for 16 weeks in the present study. What is the rationale and significance of 2/3 reduction and administration period of BCAAs? Authors should describe the circulating BCAAs levels in each model with normal range of BCAA. In addition, were the effects of BCAAs dose-dependent?

  1. The restriction of BCAAs within a high-fat diet (HFD) maintained the mice’s weight, subcutaneous and visceral adipose tissue weight. Why did the restriction of BCAAs prevent obesity? Authors should describe the data of daily food intake, calorie and locomoter activity in each model.

  1. The restriction of BCAAs prevented these metabolic dysfunctions. Authors should examine the detail mechanism of improvement of several metabolic parameters by reduction of BCAAs. Gluconeogenesis and lipid synthesis by activating fructose-1,6-bisphosphatase 1 and acetyl-CoA carboxylase alpha, acetyl-CoA from BCKA oxidation in mitochondria were actually changed in the present study?

  1. A total of 3397 individuals were included in the cohort study, 91% of the participants were Caucasian. Did not race differences influence BCAA metabolic pathway? In addition, the cohort study design does not allow for the identification of causal relationships. Authors should carefully describe the point.

  1. Restricting BCAAs per se had the strongest effect on preventing the decrease of the M2Φ level. Authors should more discuss the mechanism in greater detail

Round 2

Reviewer 3 Report

The revision has improved the manuscript. 

I have no further concern.